# Indoxacarb-Loaded Anionic Polyurethane Blend with Sodium Alginate Improves pH Sensitivity and Ecological Security for Potential Application in Agriculture

**DOI:** 10.3390/polym12051135

**Published:** 2020-05-15

**Authors:** Shiying Wang, Yi Zhang, Liupeng Yang, Qizhan Zhu, Qianli Ma, Ruifei Wang, Chaoqun Zhang, Zhixiang Zhang

**Affiliations:** 1Key Laboratory of Natural Pesticide and Chemical Biology, Ministry of Education, South China Agricultural University, Guangzhou 510642, China; hnwangshiying@163.com (S.W.); 15136283877@163.com (L.Y.); zhuqizhan@stu.scau.edu.cn (Q.Z.); mqlvincent@163.com (Q.M.); ruifeiwang@126.com (R.W.); 2Key Laboratory for Biobased Materials and Energy of Ministry of Education, College of Materials and Energy, South China Agricultural University, 483 Wushan Road, Guangzhou 510642, China; zhangyizzdx@163.com; 3Guangdong Laboratory for Lingnan Modern Agricultural Science and Technology, South China Agricultural University, 483 Wushan Road, Guangzhou 510642, China

**Keywords:** indoxacarb, anionic polyurethane, sodium alginate, nanoemulsions, bioactivity

## Abstract

Traditional pesticide formulations show poor utilization and environmental safety due to their low foliage adhesion and large auxiliaries. In this study, a novel and environment-friendly indoxacarb formulation was prepared to improve the pesticide’s utilization rate, target control characteristics and ecological security. Indoxacarb-loaded waterborne polyurethane–sodium alginate (PU/SA) nanoemulsions with film forming properties, alkaline responsive release, high effectiveness against *Spodoptera litura,* and reduced acute contact toxicity for nontarget organisms were successfully prepared. The colloidal properties, swelling and release behaviors, leaf adhesion, degradation dynamics and bioactivity assay of the indoxacarb-loaded PU/SA nanoemulsions were determined. Results showed that the obtained indoxacarb-loaded microcapsule particulates were approximately 57 nm in diameter, electronegative −45.9 mV, and uniformly dispersed in the nanoemulsions. The dried latex films of PU/SA in the alkaline environment revealed better responsive swelling and release characteristics than those in acidic and neutral conditions. Compared with a commercial emulsifiable concentrate, the indoxacarb-loaded PU/SA nanoemulsions were useful for the targeted control of *S. litura*, which have alkaline gut and showed reduced acute contact toxicity to *Harmonia axyridia*. Furthermore, the PU/SA formulation had better foliage adhesion and indicated the property of controlled-release and a persistent effect.

## 1. Introduction

*Spodoptera litura* (Lepidoptera: Noctuidae) is a harmful omnivorous agricultural pest, especially during its larval period, which causes massive crop damage and yield loss [1,2,3,4]. *S. litura* larvae have an alkaline digestive tract, and some studies have shown that it has a midgut pH > 9 [1,5]. The use of chemical pesticides, such as indoxacarb emulsifiable concentrates (ECs), via spray treatment is the most effective measure against *S. litura* [6,7]. However, the large amount and high frequency of pesticide applications may cause ecological and environmental problems due to the large organic solvent and surfactant concentrations in pesticide formulations [8]. Direct indoxacarb exposure is harmful to nontarget organisms, such as *Harmonia axyridia*, a predator of *S. litura* [9,10]. A green pesticide formulation with alkaline sensitivity and controlled release for the control of *S. litura* should be developed and prepared.

The development of intelligent microcapsules with pH-controlled release systems is a research hotspot in pesticide formulations [11,12]. A pH-controlled release formulation requires drug carriers to have a pH-responsive character, which is based on their physicochemical properties and molecular structure that allows the adjustment of states with pH environment changes [13]. Alkaline pH-responsive drug delivery systems usually have acidic groups such as microcapsules with a carboxyl group as the response factor. The side chain of the carboxyl group on the polymer segment becomes deprotonated, leading to an increase in hydrophilicity and a decrease in the hydrophobic ratio under alkaline conditions, resulting in destroyed microcapsule structure and drug release [14]. Sarkar [15] has used carboxymethyl cellulose with citric acid in the presence of bentonite to achieve the base-triggered release of an insecticide system for the efficient control of pests with alkaline guts. The pesticide achieves large release via the hydrolysis of ester cross-linkages in carriers. Xiang [16] has used a chlorpyrifos-loaded, pH-responsively controlled-release nanosystem for the targeted control of grubs to improve the utilization efficiency of pesticides.

Foliar adhesion is an important factor of pesticide formulation and affects the utilization rate of pesticides in agriculture. Nanoemulsions with good adhesion and high drug loading performance can effectively embed pesticides and may be developed and applied in agricultural production [17]. Polyurethane (PU), which has excellent mechanical and controllable degradation properties, has been developed as carrier for highly toxic antitumor drugs [18]. Vegetable oil-based waterborne PU materials have been used as biomedical compounds due to their safety, nontoxicity, good environmental safety, and renewability [19,20]. Zhang [21] has used castor oil-based waterborne PU as carrier to prepare encapsulated avermectin nanoemulsions through emulsion solvent evaporation, and has proven that avermectin-loaded PU nanoemulsions can increase foliage adhesion via hydrogen bonding between PU latex films and the hydroxyl, carboxyl, and aldehyde groups on the foliage surface after the water volatilization of water. As drug carriers, PU is biodegradable and sensitive to pH because of its carboxylic groups [22]. Dimethylol butanoic acid (DMBA), a hydrophilic chain extender, is widely used in biodegradable anionic PU preparation. Polymer blending is a useful method to improve or modify the properties of polymer materials [23]. Sodium alginate (SA) is a water-soluble natural polysaccharide-based pH-responsive polymer, typically used as a thickener and drug delivery vector in intelligent drug delivery systems in biomedical applications [24]. The pH responsiveness and viscosity of PU can be improved through blending with SA solution [25].

In this study, indoxacarb-loaded PU/SA nanoemulsions were prepared for application as an insecticide in agriculture production to control pests with alkaline guts. The colloidal properties, latex film swelling and release characteristics of the PU/SA nanoemulsion were investigated for stability evaluation. In addition, the acute toxicity to *H. axyridia*, the leaf adhesion, and indoxacarb degradation dynamics of the nanoemulsions were compared with commercial EC formulation.

## 2. Materials and Methods

### 2.1. Materials

Indoxacarb (95%) was supplied by the Chinese Academy of Agriculture Science. Castor oil 208 (CO208, OH number = 208 mg KOH/g) was provided by Fuyu Chemical Co., Ltd. (Tianjin, China). Dimethylol butanoic acid (DMBA) was purchased from Beijing Bailingwei Technology Co., Ltd. (Beijing, China). Isophorone diisocyanate (IPDI), Triethylamine (TEA) and dibutyltin dilaurate (DBTDL) were purchased from Aladdin Reagent (Shanghai, China). Chromatographic-grade acetonitrile and methanol were acquired from Meridian Medical Technologies (Saint Louis, MO, USA). Sodium alginate (SA, 98%, *M*_W_ = 12,000–40,000) was purchased from Shanghai Yuanye Bio-Technology Co., Ltd. (Shanghai, China). All other reagents and solvents were of analytical grade.

### 2.2. Preparation of Indoxacarb-Loaded PU/SA Nanoemulsions

Indoxacarb loaded PU nanoemulsions with 10 wt % drug loading capacity (LC, %) were designed and prepared using the in situ soap-free phase inverse emulsification process [26]. The PU prepolymers were prepared following the method of Liang [27]. Anionic PU (15 wt %), was prepared as shown in Figure 1. The functional ratio of IPDI–CO208–DMBA was 2.00:1.00:0.99. First, IPDI (2.47 g), castor oil 208 (3.00 g) and the chain extender DMBA (0.82 g) were mixed under mechanical stirring for 10 min at 78 °C, and the mixture was added with 0.1% DBTDL until the mixture could almost not flow freely. The mixture was added with a small amount of methyl ethyl ketone (MEK) to reduce its viscosity, allowed to react under stirring for 2 h at 78 °C, and then cooled to room temperature. The mixture was added with 0.74 mL neutralizer TEA under stirring for 30 min and slowly mixed with 35.63 mL of deionized water at a high stirring speed (600 rpm) for 2 h. MEK was completely removed though vacuum rotary evaporation to obtain the anionic PU emulsions. The indoxacarb nanoemulsions (10 wt %) prepared with 0.63 g indoxacarb dissolved in acetone solution and added before deionized water. A certain amount of the predissolved indoxacarb-acetone solution was added into the prepared anionic PU prepolymer and mixed under continuous stirring for 30 min. A predetermined amount of water was charged into the mixture with continuous stirring for 2 h. Finally, the indoxacarb-loaded PU nanoemulsions were obtained by removing the acetone with vacuum rotary evaporation at 40 °C. The LC and encapsulation efficiency (EE, %) were 9.08 ± 0.42 and 95.28 ± 0.46, and were determined using the following equations [28]: (1)$$\mathrm{LC}=\frac{M(\mathrm{indoxacarb}\;\mathrm{loaded})}{M(\mathrm{indoxacarb}\;\mathrm{loaded}\;\mathrm{PU}\;\mathrm{microcapsules})}\times 100\%.$$
(2)$$\mathrm{EE}=\frac{M(\mathrm{indoxacarb}\;\mathrm{loading})}{M(\mathrm{total}\;\mathrm{amount}\;\mathrm{of}\;\mathrm{indoxacarb})}\times 100\%.$$

An aqueous SA solution (2 wt %) was prepared by stirring preliminary swollen SA in ultrapure water for 1 h at 60 °C and removing bubbles by ultrasonic cleaning. In accordance with the designed formula, a series of PU/SA compositions (99/1, 95/5, 90/10, *v*/*v*) was prepared through the addition of SA solution with different concentrations to the prepared nanoemulsions.

### 2.3. Characterization of the Indoxacarb-Loaded PU/SA Nanoemulsions

The composition of the indoxacarb-loaded PU/SA nanoemulsions were determined qualitatively using Fourier transform infrared (FTIR) spectroscopy (Bruker Co., Hanau, Germany). The hydrodynamic particle size, particle distribution (PDI) and Zeta potential (ζ) were measured via laser scattering using the Zeta-sizer Nano ZSE (Malvern Instruments, Worcestershire, UK) at 25 °C. The morphology of the loaded microcapsules was observed using transmission electron microscopy (TEM, FEI Co., Hillsboro, OR, USA).

### 2.4. Evaluation of Swelling Properties and Indoxacarb Release In Vitro

The responsive swelling of the PU/SA nanoemulsions was defined as the weight of water absorbed per weight of dried film composition under the influence of time and pH in a swelling medium. The nanoemulsion (200 μL) was placed in a glass plate and dried to constant weight (M1) at 50 °C. The dried film was swelled in different conditions and its weight was recorded at different time points (M2). The pH-responsive swelling characteristics of gel compositions were determined by observing their swelling behavior by using deionized water at different pH (3.0, 5.0, 7.0, 9.0 and 11.0) deionized water at room temperature and weighing them at fixed time intervals. All samples were measured thrice and the swelling degree (SD) was calculated using the following equation.
(3)$$\mathrm{SD}=\frac{M2-M1}{M1}\times 100\%$$

The sensitive release behaviors of indoxacarb from the nanoemulsions were investigated in Tween/deionized water (0.1%, *v*/*v*) at different pH conditions. The release suspension (2 mL) in different pH media was sampled at certain time intervals (0, 5, 10, 20, 30, 45, 60, 120 and 240 min). The samples were analyzed at the maximum absorbance of 230 nm by using high-performance liquid chromatography (HPLC, Agilent 1260, USA) with an XDB-C18 column. The mobile phase was methanol/water (80:20, *v*/*v*) with a flow rate of 1 mL/min [29]. All treatments were performed in triplicate and the cumulative amount of indoxacarb released at different time intervals was calculated and plotted against time to study the indoxacarb release pattern.

### 2.5. Adhesion and Degradation Dynamics Studies

Given the application dose of the commercial indoxacarb EC, the indoxacarb nanoemulsions were diluted to 0.15 g/L using deionized water. Contact angle and contact diameter were detected using a contact angle measuring device (TST-U805, Climate instrument CO., Guangdong, China) following the method of Zhou [30]. Approximately 2 μL each of diluted solution and EC formulation with equal concentration were dropped on cabbage leaf surfaces. Pure water served as a reference. The changes in contact angle and evaporation over time were observed and measured.

The indoxacarb degradation dynamics in cabbage was studied via HPLC [31]. The indoxacarb nanoemulsions and EC formulations were sprayed on wild cabbage in the field to study the dissipation behavior of indoxacarb. Cabbage samples were collected at 0 (2 h), 1, 3, 5, 7, 14, 21 and 30 days after applied completion.

### 2.6. Biological Toxicity Determination

Insecticidal activity: bioassays were performed with the third instars of *S. litura* with a series of indoxacarb concentrations via the leaf dip method [32]. Mortality was investigated at 48 and 72 h. First, the clean cabbage leaves were cut into discs with 2 cm diameter by using a puncher, dipped in desired concentrations for 30 s, dried and placed in a bioassay culture dish. Finally, the equal-sized active third instars of *S. litura* were selected and placed in the bioassay dish. The test leaves were replaced with fresh cabbage leaves every 24 h [33].

An acute contact toxicity test for *H. axyridia* was established to compare the toxicity of the indoxacarb-loaded PU/SA formulation with that of the commercial EC formulation for nontarget organisms via the dry film residue method. The two indoxacarb formulations were dissolved in 1000 ug/mL stock solution with acetone-deionized water solution (50/50, *v*/*v*). The formulations were then diluted to different concentrations (200, 100, 50, 25 and 12.5 ug/mL). A barrel-shaped insect tube (3 cm × 9 cm) was used for acute contact toxicity experiments and 1 mL diluted pesticide solution was dispensed into a test tube through a transfer liquid gun. The uniform dry pesticide films were placed in the internal surface of the test tube formed through the repeated rotation and evaporation of the pesticide solution. The control was treated with acetone-deionized water solution. After drying, the third instar larva of *H. axyridia* with the same growth period was selected in each tube. Ten larvae were treated with pesticide and each treatment was repeated thrice. Enough aphids were provided to feed the H. axyridia larvae in each treatment tube. Finally, the number of dead *H. axyridia* was counted after 48 h of treatment. 

## 3. Results and Discussion

### 3.1. Preparation of Indoxacarb-Loaded PU/SA Nanoemulsions

The obtained indoxacarb-loaded PU blend with SA solution is presented in Figure 2. TEM showed that the indoxacarb-loaded PU microcapsules (Figure 2a) were spherical and had a diameter of approximately 57 nm. After blending with SA, the indoxacarb-loaded microcapsules showed increased particle size with increased SA content, whereas the PU microcapsules showed no change. With increased SA blend, the nanoemulsions appeared unstable and precipitated. Unstable pesticide nanoemulsions were not suitable for long-term storage and direct use in the field. Uniform and stable nanoemulsions allowed the spread of insecticides on the plant surface.

The colloidal properties of the indoxacarb-loaded PU/SA nanoemulsions are shown in Table 1. The hydrodynamic particle size, particle distribution (PDI) and Zeta potential (ζ) were determined to evaluate the properties of nanoemulsions, which are important for agricultural application. The uniformity of nanoemulsions result in good surface adhesion on crops and pesticide utilization for agricultural application. When the SA content was increased from 0 to 10 wt %, the microcapsule particulatesranged from 57.2 ± 0.6 nm to 70.6 ± 1.4 nm and the Zeta potential showed a slight shift in electronegativity from −45.9 ± 1.3 to −49.5 ± 0.4 mV. 

### 3.2. FTIR Analysis 

The indoxacarb-loaded PU/SA nanoemulsions were investigated using FTIR spectroscopy. The spectrum of free indoxacarb (Figure 3a) showed characteristic peaks at 3452 and 3342 cm^−1^, which were assigned to O–H and overlapping N–H stretching vibrations, respectively. The peaks at 1632, 1202, 820 and 620 cm^−1^ were due to the stretching of C=O, C−O−C (asymmetric), C−Cl and CF_3_, respectively [34]. The FTIR spectra of PU (Figure 3,) reveals the stretching vibration of N−H at 3356 cm^−1^, the characteristic stretching vibration of C=O at 1714 and 1541 cm^−1^, and the symmetric stretching vibrations of methyl groups at 2952 and 2838 cm^−1^ [35]. The SA molecule (Figure 3c) was composed of β-D-mannuronic (M) and α-L-guluronic (G) and the FTIR spectrum showed characteristic peaks at 1611, 1417 and 1031 cm^−1^, which were due to the stretching of –COO– (asymmetric), −COO– (symmetric), and C−O−C, respectively. The strong peak at 3406 cm^−1^ indicated the −OH stretching vibration from intermolecular and intermolecular hydrogen bonds. In contrast with the pure PU spectrum, the indoxacarb-loaded PU/SA spectrum showed the characteristic peaks of free indoxacarb (Figure 3d). This result proved that after blending with SA, indoxacarb did not show changes in properties and pesticide activity. In addition, given the hydrogen bonding formed between the carboxylate groups of PU and O–H of SA or the N–H of PU and carboxylate groups of SA, the characteristic peaks of the indoxacarb-loaded PU/SA broadened and moved to a high region. In general, the functional groups in nanoemulsions can increase foliage adhesion via hydrogen bonding between the nanoemulsion latex films and the hydroxyl, carboxyl and aldehyde groups on a foliage surface after water volatilization.

### 3.3. Responsive Swelling and Release Behaviors

The SD of indoxacarb-loaded PU/SA nanoemulsions film in different pH solutions is shown in Figure 4. The PU film swelling ratio increased with increasing pH because of the pH response factor, with the carboxyl group in PU appear to form COOH at the low pH condition and the carrier in a stable state due to the intermolecular hydrogen bond force. However, COOH became ionized COO− as the pH increased, and the electrostatic repulsion caused the stricture of the PU carriers to swell and become destroyed. The PU film broke after absorption for 100 min in a pH 7 solution, which may have caused the indoxacarb-loaded microcapsules to be dispersed by raindrop impact-induced erosion during agricultural application. The swelling and pH-sensitive characteristics of the PU film increased with the addition of the SA solution. The maximum SDs of the PU, PU/SA 99/1 (*v*/*v*), PU/SA 95/5 (*v*/*v*) and PU/SA 95/10 (*v*/*v*) in pH 11 solution were 200%, 250%, 280%, and 380%, respectively. Film breakage was avoided at pH 7. In addition, the cleavage of the amino group of the PU backbone under alkaline conditions can release a large amount of indoxacarb.

The pH-controlled release of indoxacarb in the indoxacarb-loaded PU and PU/SA (99/1 *v*/*v*) nanoemulsions is demonstrated in Figure 5. The indoxacarb-loaded PU nanoemulsions blend with SA solution show improved the pH-responsive release [36]. The release profiles of indoxacarb-loaded PU/SA (99:1 *v*/*v*) nanoemulsions will cause a sustained release after application in the field, and a quickly response release in the alkaline gut of pests [37,38].

### 3.4. Foliage Adhesion and Pesticide Deposition 

The initial contact angle and contact diameters of the droplets of indoxacarb-loaded nanoemulsions and EC formulation on cabbage leaf are shown in Figure 6. Given the hydrophobicity of cabbage surfaces, droplets have difficulty in forming a completely wetted state without a surface active agent [39]. The commercial indoxacarb EC formulations contain large organic auxiliaries and surfactants for good dispersion and adhesion effects on the crop surface [40]. However, traditional pesticide formulations cause environmental contamination and threaten nontarget organisms [41]. The indoxacarb-loaded nanoemulsions showed a lower contact angle (100.3°) compared with the EC formulation (132.8°) and water (134.4°). This result is due to the large functional groups of nanoemulsions forming hydrogen bonds with hydroxyl, carboxyl and aldehyde groups from the foliage surfaces [42]. The corresponding contact diameter of the nanoemulsions on the surface of cabbage leaf (1.91 mm) was higher than that of EC (1.25 mm) and water (1.37 mm) at the initial volume and concentration of 2 µL and 0.15 g/L, respectively. In addition, the enlarged evaporation area increased with the initial contact diameter of droplets, which was conducive to the formation of pesticide films on the leaf surface [43]. The corresponding dynamic changes in contact angle and droplet height in water (Figure 6b) in the indoxacarb-loaded PU/SA (99/1) nanoemulsions (Figure 6c) and indoxacarb EC formulations (Figure 6d) were detected with time to evaluate pesticide deposition.

### 3.5. Digestion Dynamics

Indoxacarb degradation was evaluated to determine the dynamics of indoxacarb after its application in the field. As shown in Figure 7, the decline curves of the indoxacarb residues corresponded with first-order kinetics. The initial concentrations of indoxacarb in cabbage 2 h after the application of the EC and PU/SA nanoemulsions were 5.02 and 6.83 mg/kg, respectively. The initial concentration of indoxacarb revealed that PU/SA nanoemulsions may have a favorable adhesion on cabbage leaf and less pesticide loss compared to the EC formulation. The corresponding half-life of indoxacarb after PU/SA nanoemulsion use was 6.03 days, which was longer than that of the EC formulation (3.96 days). Through the microencapsulation technology, the indoxacarb embedded in carrier materials displays sustained release and prolonged control efficiency [44].

### 3.6. Biological Toxicity

Insecticidal activity: The toxicity results of the indoxacarb-loaded PU/SA nanoemulsions against S. litura are shown in Table 2. The excellent pH-sensitive ability of indoxacarb loaded PU/SA nanoemulsions showed favorable insecticidal activity against *S. litura*. The LC_50_ was used to evaluate the toxicity of the nanoemulsions. After 48 and 72 h, the LC_50_ values of the nanoemulsions were 5.58 and 1.55 mg/kg, respectively, which were better than those of EC formulation (6.69 and 1.71 mg/kg, respectively). The unloaded PU/SA nanoemulsion treatment (control) was nontoxic to *S. litura*. Results indicated that the indoxacarb microcapsules achieved structural transformation, chain extension, and rapid indoxacarb release in the alkaline midgut of *S. litura* [45]. In addition, microencapsulation extended the surface area of indoxacarb in the gut of *S. litura* and caused increased contact with the target area.

Acute contact toxicity of *H. axyridia*: Figure 8 reveals the toxicity of the indoxacarb-loaded PU/SA nanoemulsions to *H. axyridia* compared with that of the EC formulation by using the dry film residue method at serious concentration. Results showed that the commercial EC formulation showed high toxicity to the third instar larva of *H. axyridia*, and the mortality rate ranged from 43% to 100% at the effective concentration from 12.25 to 200 ug/mL. However, the toxicity of the indoxacarb-loaded PU/SA nanoemulsions to *H. axyridia* relatively decreased under the same concentration because the indoxacarb microencapsulation reduced acute contact. The acute contact toxicity assays showed that the indoxacarb-loaded PU/SA nanoemulsions were nontoxic at effective concentrations of 12.5, 20 and 50 ug/mL, whereas the EC formulation demonstrated high toxicity. Therefore, the indoxacarb-loaded PU/SA nanoemulsions were safe for nontarget organisms and ensured pest control efficacy in agricultural applications. 

## 4. Conclusions

The indoxacarb-loaded PU/SA nanoemulsions easily adhere to and spread on plant surfaces without a high amount of additives, and are rapidly deposited as pesticide films, which are beneficial for spray application in agriculture production. The nanoemulsion is a water-based eco-friendly pesticide formulation, consisting of nontoxic and renewable materials as carriers to encapsulate indoxacarb, effectively reducing the environmental harm of pesticides. The indoxacarb-loaded PU/SA nanoemulsion films have shown a favorable alkaline pH response swelling and release capability result in a better target control of *S. litura*, meanwhile reduce the acute contact toxicity of *H. axyridia* via microencapsulation. In particular, the optimized PU/SA (99/1, *v*/*v*) nanoemulsions have shown better foliar adhesion and deposition effects than the commercial preparation. Thus, the nanoemulsions can improve the utilization rate of pesticides. In addition, the sustained release of indoxacarb provides a persistent pest control effect and is beneficial in reducing the frequency and dosage of pesticides.

## Figures and Tables

**Figure 1 polymers-12-01135-f001:**
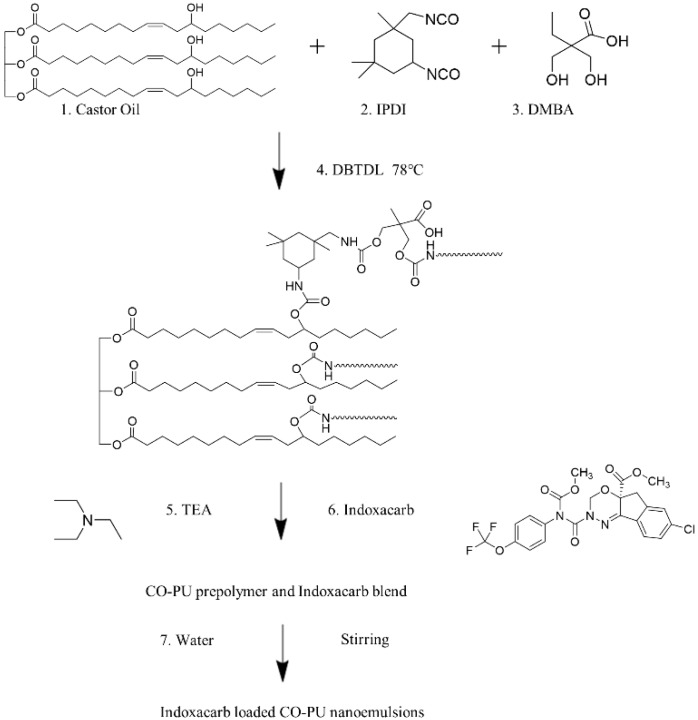
Synthesis route of the Indoxacarb loaded anionic waterborne CO-PU nanoemulsions.

**Figure 2 polymers-12-01135-f002:**
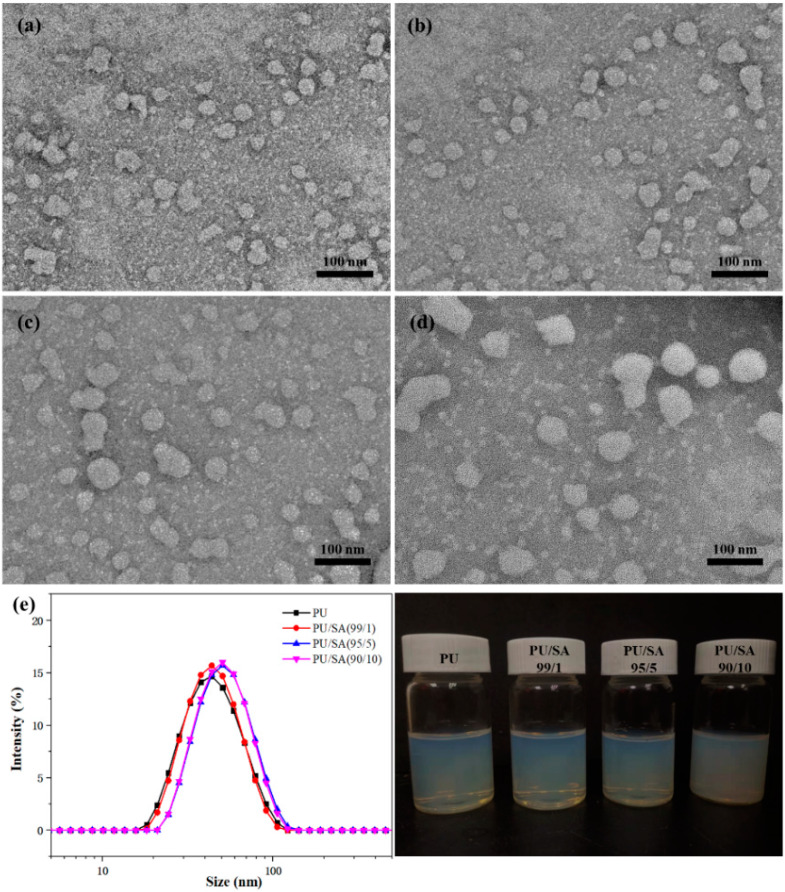
TEM micrographs of the indoxacarb-loaded PU/SA nanoemulsions: (**a**) PU; (**b**) PU/SA (99/1); (**c**) (95/5); (**d**) (90/10); and the corresponding particle size distribution (**e**).

**Figure 3 polymers-12-01135-f003:**
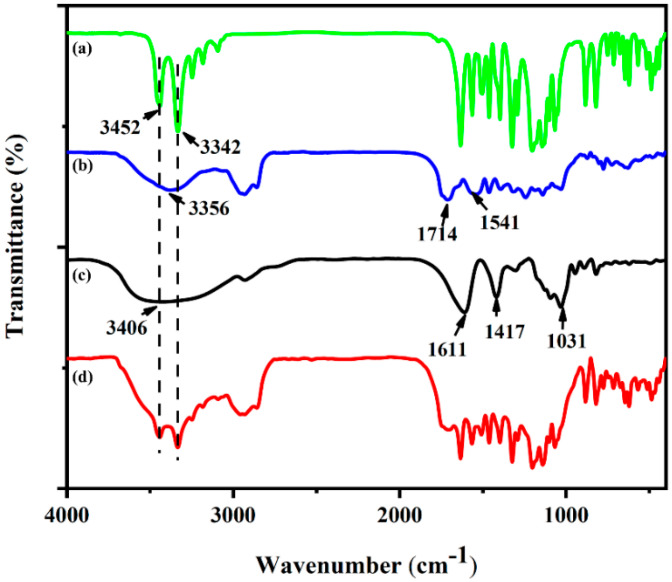
FTIR analysis of indoxacarb (**a**); PU (**b**); SA (**c**) and indoxacarb-loaded PU/SA nanoemulsions (**d**).

**Figure 4 polymers-12-01135-f004:**
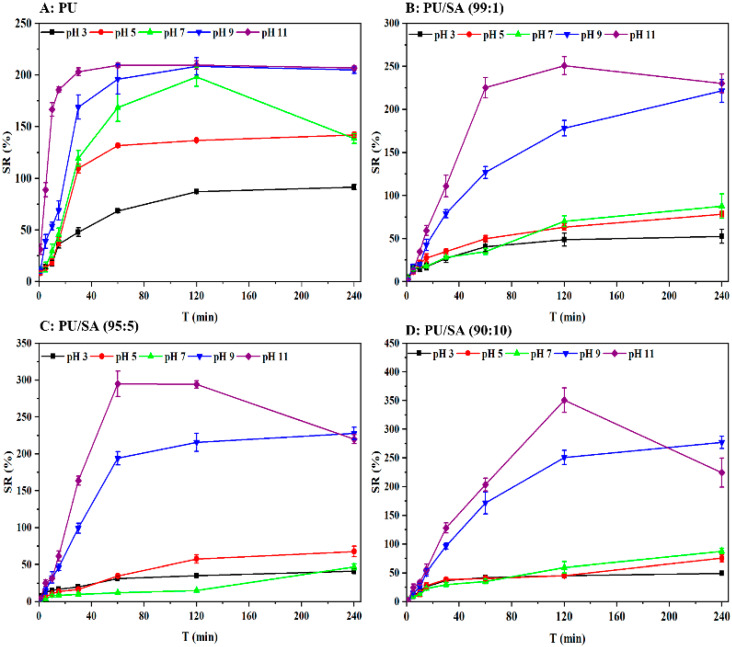
Swelling degree of PU (**A**); PU/SA (99/1, **B**); PU/SA (95/5, **C**) and PU/SA (90/10, **D**) gel compositions in different pH solution. All data represent means ± SE (n = 3).

**Figure 5 polymers-12-01135-f005:**
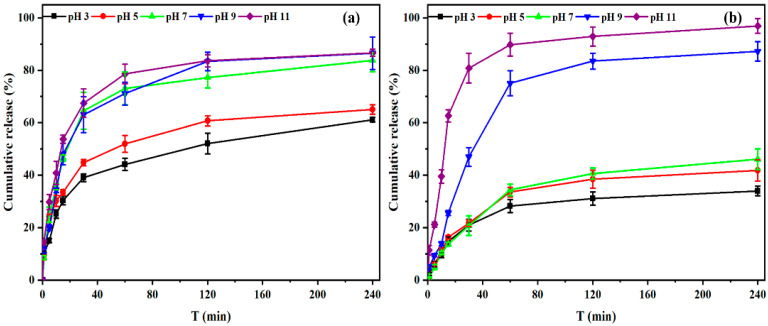
Release profiles of the indoxacarb-loaded PU (**a**) and PU/SA (99/1, **b**) in different pH solution. All data represent means ± SE (n = 3).

**Figure 6 polymers-12-01135-f006:**
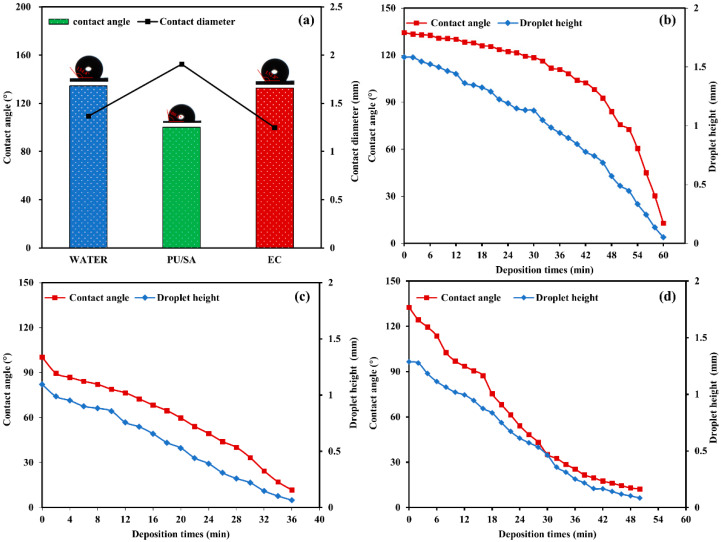
Contact angle and contact diameter of cabbage leaves treated by different indoxacarb formulations (PU/SA (99/1) and EC) with the initial volume of 2 uL (**a**) in a certain concentration (0.15 g/L), water sample as control contrast. The corresponding dynamic change of contact angle and droplet height in Water (**b**), PU/SA (99/1) (**c**) and EC (**d**).

**Figure 7 polymers-12-01135-f007:**
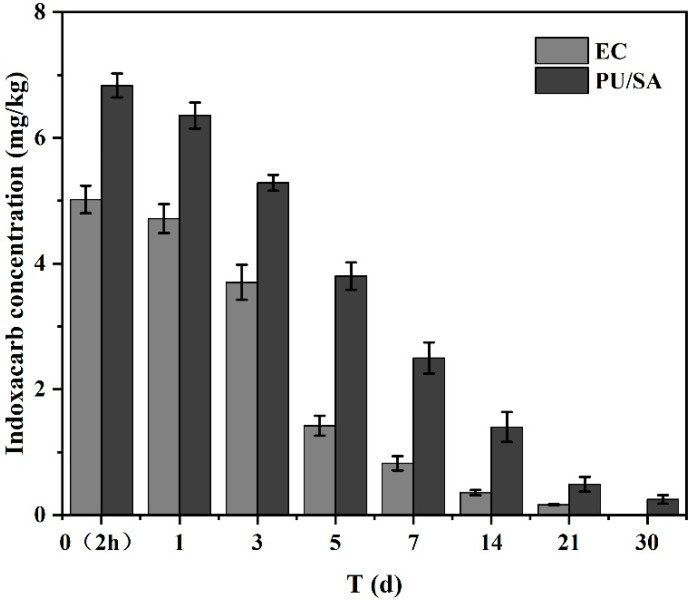
Degradation studies of indoxacarb in cabbage by treatment with PU/SA nanoemulsions and EC formulation.

**Figure 8 polymers-12-01135-f008:**
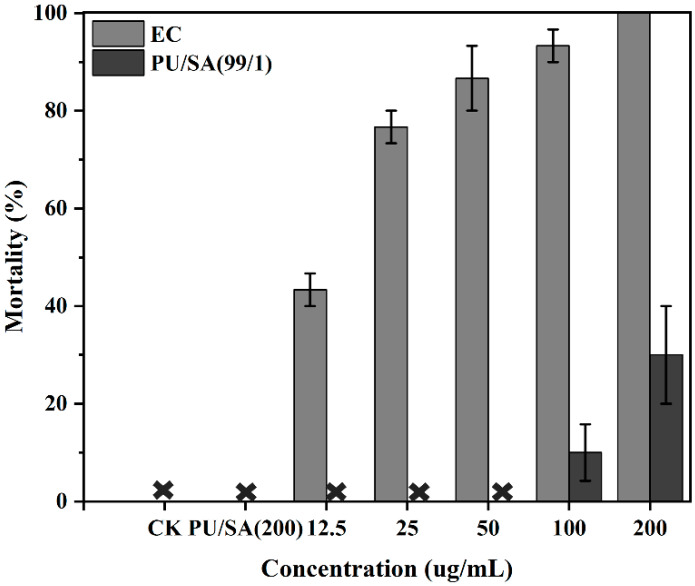
Acute contact toxicity test to *H. axyridia* treated with indoxacarb-loaded PU/SA nanoemulsions and EC formulation by a dry film residue method. The effective concentration from 12.25 ug/mL to 200 ug/mL and the morality was investigated at 48 h. The shape of “
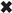
” indicate nontoxic at the corresponding treatment.

**Table 1 polymers-12-01135-t001:** Colloidal properties of indoxacarb-loaded PU/SA nanoemulsions.

Sample	Size (nm)	DPI	ζ (mV)
PU	57.2 ± 0.6	0.113	−45.9 ± 1.3
PU/SA (99/1)	58.4 ± 0.3	0.114	−47.3 ± 0.3
PU/SA (95/5)	60.1 ± 1.9	0.133	−48.8 ± 0.6
PU/SA (90/10)	70.6 ± 1.4	0.196	−49.5 ± 0.4

Note: All data represent means ± SE (n = 3). The hydrodynamic particle size, particle distribution (PDI) and Zeta potential (ζ) were measured with laser scatter method by a Zeta-sizer Nano ZSE (Malvern Instruments, UK) at 25 °C.

**Table 2 polymers-12-01135-t002:** Insecticidal activity of indoxacarb-loaded PU/SA nanoemulsions against *S. litura*.

Treatment		Regression Equation	LC_50_ (mg/kg)	95% Fiducial Limit	Correlation Coefficients
Indoxacarb	48 h	*y* = 4.0116 + 1.1970*x*	6.69	4.07–11.00	0.9923
	72 h	*y* =4.5125 + 2.0823*x*	1.71	1.30–2.26	0.9855
Indoxacarb-Loaded PU/SA (99/1)	48 h	*y* = 4.0116 + 1.1970*x*	5.58	3.42–9.08	0.9923
	72 h	*y* = 4.6028 + 2.0791*x*	1.55	1.17–2.07	0.9903

Note: The insecticidal activity test via a leaf dip method reared at 25 ± 2 °C and 65% ± 5% RH condition.

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
