# Peer review of "Indoxacarb-Loaded Anionic Polyurethane Blend with Sodium Alginate Improves pH Sensitivity and Ecological Security for Potential Application in Agriculture"

_polymers, 2020, doi:10.3390/polym12051135_

Round 1

Reviewer 1 Report

The manuscript is well written but needs major English corrections.
In addition, in order to be considered for publication, additional cytotoxicity tests are needed.
We cannot recommend materials for being used without knowing if they pose any toxic effects towards human organism.

Author Response

1. The reviewer’s comment:The manuscript is well written but needs major English corrections.

The authors’ Answer: Thank you for your comments concerning our manuscript.  The manuscript has been revised, accordingly. The manuscript has been revised thoroughly by a Native-English speaker.

2.  In addition, in order to be considered for publication, additional cytotoxicity tests are needed.We cannot recommend materials for being used without knowing if they pose any toxic effects towards human organism.

The authors’ Answer: In this work, the castor oil-based anionic waterborne polyurethane has been developed as carriers for indoxacarb. And this vegetable oil-based waterborne polyurethanes have been used as biomedical materials in our previous studies due to their safety, nontoxicity, good environmental safety, and renewability [1-3].

1. Zhang Y, He W, Li J, Wang K, Li J, Tan H, Fu Q. Gemini quaternary ammonium salt waterborne biodegradable polyurethanes with antibacterial and biocompatible properties. Materials Chemistry Frontiers 2017, 1(2): 361-368.

2. Zhang Y, He X, Ding M, He W, Li J, Li J, Tan H. Antibacterial and Biocompatible Cross-Linked Waterborne Polyurethanes Containing Gemini Quaternary Ammonium Salts. Biomacromolecules 2018, 19(2): 279-287.

3.  Ding M, Li J, He X, Song N, Tan H, Zhang Y, Zhou L, Gu Q, Deng H, Fu Q. Molecular Engineered Super-Nanodevices: Smart and Safe Delivery of Potent Drugs into Tumors. Adv Mater 2012.

Reviewer 2 Report

Manuscript entitled “Indoxacarb loaded anionic polyurethane blend with sodium alginate improves pH sensitivity and ecological security for potential application in agriculture” submitted by Shiying Wang, Yi Zhang, Liupeng Yang, Qizhan Zhu, Qianli Ma, Ruifei Wang, Dongmei Cheng, Chaoqun Zhang, Zhixiang Zhang, can be accepted for publishing in Polymers Journal, after major revision.

In this study, indoxacarb loaded PU/SA nanoemulsions was prepared and used as a potential insecticide in agriculture production for control alkaline gut pests. The manuscript presents original results, obtained in relatively well organized and systematic way, but the interpretation of the experimental results is quite poor.

Here is a list of my specific comments:

  1. General comment: The novelty and practical applicability of this study should be highlighted more.
  2. Page 1, 1. Introduction: This section is too brief and should be detailed in order to describe the state of art in this field.
  3. Page 2, 2. Materials and Method: This section should be reorganized. Pay attention on the technical details important to describe the experimental methodology and analysis methods used for this study.
  4. Page 2, line 81: Replace “ultrasonic cleaning machine” with “ultrasonic cleaning”.
  5. Page 4, 3. Results: Replace this title by “Results and discussion” and provide a detailed interpretation of all the results included in this section.
  6. Page 4, lines 137-146: The paragraphs: “Anionic polyurethane (PU, 15 wt. %) was made with… were detected with 95.28 ± 0.46 and 9.08 ± 0.42 respectively.” should be moved in Experimental section.
  7. Page 4, 3.1. Preparation of indoxacarb loaded PU/SA nanoemulsions: What is the relevance of these results?? Why are important these characteristics?? A more detailed interpretation of these properties should be added here.
  8. Page 6, 3.2. FTIR analysis: The same observation as above.
  9. Page 10, 4. Conclusions: This section is too brief and should be detailed. Include here the most important experimental results of this study.

Author Response

1.The reviewer’s comment: General comment: The novelty and practical applicability of this study should be highlighted more.

The authors’ Answer: We have perfected and emphasized the novelty and practical applicability of this study. The indoxacarb loaded PU/SA nanoemulsions is a water-based eco-friendly pesticide formulation, use non-toxic and renewable materials as carrier materials to encapsulated indoxacarb, can effectively reduce the harm of pesticides to the ecological environment. The optimized PU/SA (99/1, v/v) nanoemulsions shows better foliar adhesion and deposition effect than the commercial preparation, which will improve the utilization rate of pesticides. In addition, the indoxacarb film of PU/SA nanoemulsions shown a favorable alkaline pH response swelling and release capability result in a better target control of S. litura, meanwhile reduce the acute contact toxicity of H. axyridia.

2.The reviewer’s comment:Page 1, 1. Introduction: This section is too brief and should be detailed in order to describe the state of art in this field.

The authors’ Answer: We have perfected and supplemented the research content of pH-responsive drug release system and vegetable oil-based polyurethane as drug carrier technology in the introduction.

3.The reviewer’s comment:Page 2, 2. Materials and Method: This section should be reorganized. Pay attention on the technical details important to describe the experimental methodology and analysis methods used for this study.

The authors’ Answer: We have reorganized and modified the content of the Materials and Method section, and improved the test and analysis method.

4.The reviewer’s comment:The reviewer’s comment:Page 2, line 81: Replace “ultrasonic cleaning machine” with “ultrasonic cleaning”.

The authors’ Answer: The spelling and syntax errors have been checked and corrected.

5.The reviewer’s comment:Page 4, 3. Results: Replace this title by “Results and discussion” and provide a detailed interpretation of all the results included in this section.

The authors’ Answer: The title was replaced with “Results and discussion”. In addition, the description of the results section has been refined and supplemented.

6.The reviewer’s comment:Page 4, lines 137-146: The paragraphs: “Anionic polyurethane (PU, 15 wt. %) was made with… were detected with 95.28 ± 0.46 and 9.08 ± 0.42 respectively.” should be moved in Experimental section.

The authors’ Answer: This part of the content has been transferred to the Materials and Methods section and has been modified and adjusted.

7.The reviewer’s comment: Page 4, 3.1. Preparation of indoxacarb loaded PU/SA nanoemulsions: What is the relevance of these results?? Why are important these characteristics?? A more detailed interpretation of these properties should be added here.

The authors’ Answer: We have organized and modified this part of the content. The colloidal properties and stability of the nanoemulsion were determined by electron microscopy and particle size and distribution analysis. The characteristics of the colloid is crucial for the application of pesticide formulations in agriculture. The more uniformly of nanoemulsions the more well surface adhesion on crops and pesticide utilization in agricultural application. The uniform and stable nanoemulsions helps spread on plant surface and then enhance the pesticide utilization.

8.The reviewer’s comment:Page 6, 3.2. FTIR analysis: The same observation as above.

The authors’ Answer: FTIR analysis results indicate that after blend with SA, the properties of indoxacarb have not changed and will not affect pesticide activity. In addition, due to the hydrogen bonding formed between carboxylate groups of PU and O–H of SA or N–H of PU and carboxylate groups of SA, the characteristic peaks of indoxacarb loaded PU/SA will broadening and move to the higher region. And these functional group in the nanoemulsions could increase foliage adhesion via hydrogen bonding between nanoemulsions latex films and hydroxyl, carboxyl and aldehyde groups on the foliage surface after the volatilization of water.

9.The reviewer’s comment:Page 10, 4. Conclusions: This section is too brief and should be detailed. Include here the most important experimental results of this study.

The authors’ Answer: We have replenished and improved the content of the Conclusions section.

Reviewer 3 Report

This work presented a proper way to prepare the novel ecological materials. The comments of the reviewer are showing below,

1. The authors should introduce more examples of polymeric ecological materials in the introduction section.

2. What is the reason that the maximum SA content is about 10 in the nanoemulsions? Is there any possibility to increase the SA content in the nanoemulsions?

Author Response

1.The reviewer’s comment: The authors should introduce more examples of polymeric ecological materials in the introduction section.  

The authors’ Answer: We have perfected and supplemented the research content of pH-responsive drug release system and vegetable oil-based polyurethane as drug carrier technology in the introduction.

2.The reviewer’s comment: What is the reason that the maximum SA content is about 10 in the nanoemulsions? Is there any possibility to increase the SA content in the nanoemulsions?

The authors’ Answer: By blend with SA solution, the pH-responsive character and viscosity of PU can be improved. However, with the increase of SA blend, the nanoemulsions will appear unstable and cause precipitation. Unstable pesticide nanoemulsions will not be suitable for long-term storage and directly used in the field. The uniform and stable nanoemulsions helps spread on plant surface and enhance foliage adhesion.

Round 2

Reviewer 1 Report

The authors have considered all the comments in the revised version.

Reviewer 2 Report

All my previous remarks and comments have been considered into new version of the manuscript. It means that reviewed manuscript meets the criteria and in my opinion can be published as original paper in Polymers Journal.